

# Shielding effects of myelin sheath on axolemma depolarization under transverse electric field stimulation

Hui Ye and Jeffrey Ng

Department of Biology, Loyola University of Chicago, Chicago, IL, USA

## ABSTRACT

Axonal stimulation with electric currents is an effective method for controlling neural activity. An electric field parallel to the axon is widely accepted as the predominant component in the activation of an axon. However, recent studies indicate that the transverse component to the axolemma is also effective in depolarizing the axon. To quantitatively investigate the amount of axolemma polarization induced by a transverse electric field, we computed the transmembrane potential ($Vm$) for a conductive body that represents an unmyelinated axon (or the bare axon between the myelin sheath in a myelinated axon). We also computed the transmembrane potential of the sheath-covered axonal segment in a myelinated axon. We then systematically analyzed the biophysical factors that affect axonal polarization under transverse electric stimulation for both the bare and sheath-covered axons. Geometrical patterns of polarization of both axon types were dependent on field properties (magnitude and field orientation to the axon). Polarization of both axons was also dependent on their axolemma radii and electrical conductivities. The myelin provided a significant "shielding effect" against the transverse electric fields, preventing excessive axolemma depolarization. Demyelination could allow for prominent axolemma depolarization in the transverse electric field, via a significant increase in myelin conductivity. This shifts the voltage drop of the myelin sheath to the axolemma. Pathological changes at a cellular level should be considered when electric fields are used for the treatment of demyelination diseases. The calculated term for membrane polarization ($Vm$) could be used to modify the current cable equation that describes axon excitation by an external electric field to account for the activating effects of both parallel and transverse fields surrounding the target axon.

## INTRODUCTION

Electrical stimulation of nerve cells was first reported by Luigi Galvani in 1780 (*Galvani, 1791*), who accidently found that muscles from a dead frog would twitch when touched with a charged metal scalpel, a discovery that sparked the appreciation of electricity in relation to animation—or life. Today, electric stimulation of neurons in the peripheral or central nervous systems have been widely utilized for controlling neural

Corresponding author
Hui Ye, hye1@luc.edu

network activity (*Selimbeyoglu & Parvizi, 2010*), synaptic transmission (*Nowak & Bullier, 1998*), and pain (*Coderre et al., 1993*). Electric currents can also be generated via magneto-electric induction with magnetic coils for non-invasive control of neural activity (*Maccabee et al., 1991*, *1993*; *Ye et al., 2010*, *2011*; *Ye & Steiger, 2015*).

An electric field surrounding a straight nerve axon can be separated into two components: one parallel to ($E_{//}$) and the other perpendicular (traversal, $E_{\perp}$) to the axon. The $E_{//}$ is widely regarded as the predominant factor that activates the axons (*Basser, Wijesinghe & Roth, 1992*; *Roth & Basser, 1990*), which is supported by numerous experimental results (*Amassian, Maccabee & Cracco, 1989*; *Basser & Roth, 2000*). Consequently, theoretical analyses of electrical activation have predominately been focused on computing $E_{//}$ along a fiber (*Esselle & Stuchly, 1994*, *1995*; *Nagarajan & Durand, 1995*; *Ravazzani et al., 1996*; *Roth & Basser, 1990*; *Roth et al., 1990*). The current cable equation, $\lambda^2 \frac{\partial^2 \phi_m}{\partial x^2} - \tau \frac{\partial \phi_m}{\partial t} - \phi_m = -\lambda^2 \frac{\partial V_e}{\partial x^2}$, which describes axonal activation, contains only the axial term ($E_{//}$). Here, $\lambda^2 = \frac{R_m c}{2R_i}$ and $\tau = R_m C_m$ are the space and time constants, respectively. $\phi_m$ is the transmembrane potential and $V_e$ is the extracellular electric field applied to the fiber. It is nonzero only if $E_{//}$ is nonzero. The surface resistance and capacitance of the membrane are $R_m$ and $C_m$, respectively. The intracellular resistivity is $R_i$ and the fiber radius is $c$. This simplification facilities the rapid calculation of neural activation. However, it ignores the presence of the cell, which perturbs the local extracellular electric field. It also ignores the mutual interactions between the neurons and the applied electric field (*Ye & Steiger, 2015*), mainly the electric field that directly penetrates and depolarizes the cell membrane, or $E_{\perp}$.

## Transversal field for membrane polarization

Mounting evidence from experimental and simulation studies support the notion that cell membranes can be polarized by transversal electric fields. An electric field that penetrates the cell membrane was directly observed to cause polarization in hippocampal neurons (*Bikson et al., 2001*), in neural stem cells (*Zhao et al., 2015*), and in oocytes (*Lee & Grill, 2005*). When $E_{\perp}$ is extremely strong, it can even cause membrane instability and pore formation (*Bingham, Olmsted & Smye, 2010*). Analytical computations of the transverse membrane potential under electric stimulation started as early as the 1950s (*Fricke, 1953*; *Schwan, 1957*) for a simple cellular shape. Recent works have calculated membrane polarization by the transverse field in cells with more complex geometry (*Kotnik & Miklavcic, 2000a*, *2000b*), and by the transverse electric field induced by time-varying magnetic field (*Ye, Cotic & Carlen, 2007*; *Ye et al., 2011*).

Because of the observable, polarizing effects of the transverse field on large structures like the cell body, it is reasonable to speculate that a transverse field could also play a significant role in the polarization of axons. Indeed, evidence favoring transversal activation of axons also appear in the literature. It was reported (*Pourtaheri et al., 2009*) that individual axons can be selectively activated by a transverse field in a nerve bundle. These fields produce strong effects in the stimulation of ulnar nerves (*Cros, Day & Shahani, 1990*; *Olney et al., 1990*) and long fibers (*Grill & Wei, 2009*). Using a magnetic

coil to induce the electric field, *Ruohonen et al. (1996)* discovered that activation of peripheral nerves could occur when the coil was oriented in a way that only generated $E_\perp$, and a later theoretical work (*Ye et al., 2011*) confirmed the axonal depolarization by this field. Clinically, the fast switching of magnetic fields during magnetic resonance imaging (MRI) scanning generates $E_\perp$ in patients, which is considered an important risk factor for unwanted peripheral nerve stimulation (*While & Forbes, 2004*).

At present, the consensus is that $E_\perp$ is a modulator to the dominant effects caused by $E_{//}$, although some researchers have speculated that the stimulation effects from transverse fields may arise due to nerve undulation, which generates longitudinal field components (*Lontis, Nielsen & Struijk, 2009*; *Schnabel & Struijk, 1999*; *Struijk & Durand, 1998*). In the presence of $E_{//}$, it was thought that $E_\perp$ could introduce subthreshold membrane depolarization which enhances stimulation by providing an additive effect on $E_{//}$ (*Lontis, Nielsen & Struijk, 2009*). Alternatively, $E_\perp$ may provide rapid axonal polarization in the transverse direction and $E_{//}$ drives the slow development of the mean transmembrane potential (*Cranford, Kim & Neu, 2012*). $E_{//}$ and $E_\perp$ could potentially provide a strategy for differential activation of axons with different properties (*Ruohonen et al., 1996*).

## Modify cable equation to include the contribution of transverse field

In cases with significant transverse stimuli, where the membrane-field interaction is sufficient and polarization is primarily due to the transverse field, the cable model assumptions are known to be invalid (*Krassowska & Neu, 1994*). Many modeling studies have argued for the inclusion of the transverse field for the accurate simulation of neural activation, as well as the development of mathematical tools to serve this purpose (*Gimsa & Wachner, 2001*; *Kotnik & Miklavcic, 2000a*, *2006*; *Krassowska & Neu, 1994*; *Ye et al., 2011*; *Ye & Steiger, 2015*).

Several papers have reported their first endeavors for modifying the cable equation. *Yu, Zheng & Wang (2005)* modified the activation function to include the transversal field in magnetic stimulation. *Ravazzani et al. (1996)* magnetically stimulated the median nerve and recorded the evoked muscle responses, and discovered that including the transversal field in the cable equation provided a much improved correlation between the mush Electromyography (EMG) and the activating function. A recent endeavor, which modified the current cable equation to include the transversal term (*Wang et al., 2018*), showed that the transversal field could affect threshold of demyelinated axons, but not in myelinated axons. Another work by the same group included the transverse term in the cable equation that describes magnetic field stimulation (*Wang, Grill & Peterchev, 2018*). In both these studies, the membrane was represented by a resistor in parallel with a membrane capacitance. For computational simplicity, all the above-mentioned works ignored the physical presence of the lipid bilayer membrane, a "shell" like structure that has non-zero thickness. Consequently, the field perturbation caused by the membrane, which is essential for the re-distribution of the transverse field proximal to the axon (*Farkas, Korenstein & Malkin, 1984*; *Jerry, Popel & Brownell, 1996*; *Lee & Grill, 2005*; *Mossop et al., 2007*), as well as the buildup of transmembrane potential
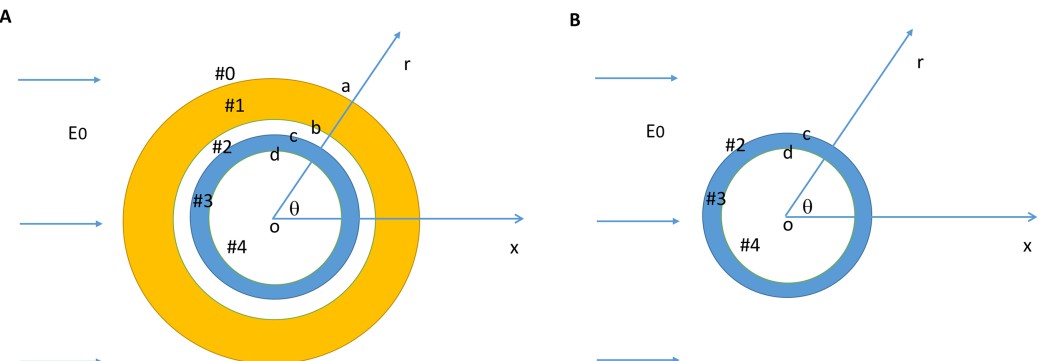

**Figure 1 Model setup for a myelin-covered axon (A) and a bare axon (B) under transverse electric field stimulation.** The cylindrical coordinate system that was used to define the orientation of the electric field and the axon.

(*Kotnik & Miklavcic, 2000a*, *2000b*) through cell-field interaction (*Ye & Steiger, 2015*), were ignored. Furthermore, in the myelinated axon model, the possibility that the presence of the myelin sheath might shield the internal structure such as the axon membrane (*Kotnik & Miklavcic, 2006*; *Ye & Curcuru, 2016*) was not considered.

In the present paper, we model a bare axon as a conductive cylindrical shell, and provide an analytical expression of membrane polarization ($Vm$) for an unmyelinated axon (or the node of Ranvier in myelinated axon), under transverse electric field stimulation. We also provide $Vm$ for a myelin-covered axon, which was modeled as a co-centric, two shell structure. We investigate biophysical factors that may affect the $Vm$, including field intensity and orientation, axonal biophysics and the impact of pathological demyelination. We discuss the possibility of placing $Vm$ into a modified cable equation, so that the contributions of both the longitudinal and transverse components of an electrical field could be simultaneously evaluated during electric stimulation of axons.

## METHODS

### Cylindrical axon model in a transverse electric field

We modeled a bare axon and its myelin sheath using homogeneous cylindrical volume conductors (*Esselle & Stuchly, 1994*; *Schnabel & Struijk, 2001*). Figure 1A illustrates the location and orientation of a myelin-covered axon in a cylindrical coordinate system ($r$, $\theta$, $z$). The axon was exposed to a transverse, direct current (DC) electric field ($E_0$). It included a total of five isotropic and homogenous regions: the medium (#0), the myelin sheath (#1), the periaxonal space (#2), the axolemma (#3), and the axonal cytoplasm (#4). The dielectric permittivities and conductivities in the five regions were $\varepsilon_0$, $\varepsilon_1$, $\varepsilon_2$, $\varepsilon_3$, $\varepsilon_4$ and $\sigma_0$, $\sigma_1$, $\sigma_2$, $\sigma_3$, $\sigma_4$, respectively. The myelin had an outer radius ($a$), inner radius ($b$), and the thickness ($b$–$a$). The axon had an outer radius ($c$) and inner radius ($d$). Thickness of the axolemma was therefore $c$–$d$. Figure 1B illustrates a bare axon, which was composed of only the periaxonal space (#2), the axolemma (#3), and the axonal cytoplasm regions.

## Governing equation and boundary conditions

Using the cylindrical coordinates $(r, \theta, z)$, the electric field distribution was calculated by

$$E = -\nabla V = -\left(\frac{\partial V}{\partial r}, \frac{1}{r}\frac{\partial V}{\partial \theta}, \frac{\partial V}{\partial z}\right) \tag{1}$$

For the DC electric field stimulation, an electric potential was obtained by solving Laplace's equation

$$\nabla^2 V = 0 \tag{2}$$

The potential, $V$, is the electric scalar potential due to the charge accumulation between the interface of the two different media (*Stratton, 1941*). In a cylindrical coordinate system $(r, \theta, z)$, it is written as

$$\frac{1}{r}\frac{\partial}{\partial r}\left(r\frac{\partial V}{\partial r}\right) + \frac{1}{r^2}\frac{\partial^2 V}{\partial \theta^2} = 0 \tag{3}$$

Several boundary conditions were evaluated in solving the equation (Appendix): (A) The electric potential was continuous across the boundary of the two different media. (B) The normal current density was continuous across the two different media. "Complex conductivity," defined as $S = \sigma + j\omega\varepsilon$, was calculated to account for the dielectric permittivity of the material (*Kotnik & Miklavcic, 2000b*; *Kotnik, Miklavcic & Slivnik, 1998*; *Polk & Song, 1990*). Here, $\sigma$ was the conductivity of the tissue, $\varepsilon$ was the permittivity, $\omega$ was the angular frequency of the field (zero for DC electric field) and $j = \sqrt{-1}$ was the imaginary unit. On the extracellular media/myelin interface (#0#1, $r = a$),

$$S_0 E_{0r} - S_1 E_{1r} = 0 \tag{4}$$

On the myelin/periaxonal interface (#1#2, $r = b$),

$$S_1 E_{1r} - -S_2 E_{2r} = 0 \tag{5}$$

On the periaxonal/axon interface (#2#3, $r = c$),

$$S_2 E_{2r} - S_3 E_{3r} = 0 \tag{6}$$

On the axon/cytoplasm interface (#3#4, $r = d$),

$$S_3 E_{3r} - S_4 E_{4r} = 0 \tag{7}$$

where $S_0 = \sigma_0 + j\omega\varepsilon_0, S_1 = \sigma_1 + j\omega\varepsilon_1, S_2 = \sigma_2 + j\omega\varepsilon_2, S_3 = \sigma_3 + j\omega\varepsilon_3, S_4 = \sigma_4 + j\omega\varepsilon_4$. (C) Electric fields an infinite distance away should not be perturbed by presence of the axon. (D) The electric potential inside the cytoplasm ($r = 0$) was finite.

## Model parameters

Table 1 lists the default values of the model parameters and their ranges. The choice of the electric parameters were based on reports in the literature (*Kotnik, Bobanovic & Miklavcic, 1997*; *Kotnik & Miklavcic, 2006*). Axon radius was selected from *Berthold & Rydmark (1995)*. The diameter of the unmyelinated axons ranges from approximately 0.1–2 μm (three μm in humans). We used 0.6 μm as the standard value and 0.1–3 μm as the range. The thickness of the axonal membrane was selected from *Nagarajan & Durand (1995)*. The diameter of the myelin was set to double the axon diameter.
**Table 1 Model parameters.**

| Parameters | Standard value | Lower limit | Upper limit |
|---|---|---|---|
| Extracellular conductivity ($\sigma_0$, S/m)[a,b] | 0.2 | $5 \times 10^{-4}$ | 2.0 |
| Myelin conductivity ($\sigma_1$, S/m)[a,b] | $5.0 \times 10^{-7}$/n[g] | $1.0 \times 10^{-8}$/n | $1.2 \times 10^{-6}$/n |
| Periaxonal conductivity ($\sigma_2$, S/m)[a,b] | 0.2 | $2.0 \times 10^{-2}$ | 1.0 |
| Axonal conductivity ($\sigma_3$, S/m)[a,b] | $5.0 \times 10^{-7}$ | $1.0 \times 10^{-8}$ | $1.2 \times 10^{-6}$ |
| Cytoplasmic conductivity ($\sigma_4$, S/m)[a,b] | 0.2 | $2.0 \times 10^{-2}$ | 1.0 |
| Extracellular dielectric permittivity ($\varepsilon_0$, As/$Vm$)[a,b] | $6.4 \times 10^{-10}$ | $3.5 \times 10^{-10}$ | $7.0 \times 10^{-10}$ |
| Myelin dielectric permittivity ($\varepsilon_1$, As/$Vm$)[a,b] | $4.4 \times 10^{-11}$ | $1.8 \times 10^{-11}$ | $8.8 \times 10^{-11}$ |
| Periaxonal dielectric permittivity ($\varepsilon_2$, As/$Vm$)[a,b] | $6.4 \times 10^{-10}$ | $3.5 \times 10^{-10}$ | $7.0 \times 10^{-10}$ |
| Axonal myelin dielectric permittivity ($\varepsilon_3$, As/$Vm$)[a,b] | $4.4 \times 10^{-11}$ | $1.8 \times 10^{-11}$ | $8.8 \times 10^{-11}$ |
| Cytoplasmic dielectric permittivity ($\varepsilon_4$, As/$Vm$)[a,b] | $6.4 \times 10^{-10}$ | $3.5 \times 10^{-10}$ | $7.0 \times 10^{-10}$ |
| Myelin diameter (a, $nm$) | 1.5 | 0.7 | 4.6 |
| b. Axonal membrane thickness ($nm$)[c] | 6 | 4 | 8 |
| Axonal radius (c, μm)[d] | 0.6 | 0.1 | 1.2 |
| Periaxonal space width (μm)[e] | 0.004 | 0.004 | 0.004 |
| Number of myelin layers ($n$)[f] | 40 | 0 | 40 |
| Electric field intensity (V/m) | 200 | 0 | 200,000[h] |

Notes:
 [a] *Kotnik, Bobanovic & Miklavcic (1997).*
 [b] *Kotnik & Miklavcic (2006).*
 [c] *Nagarajan & Durand (1995).*
 [d] *Berthold & Rydmark (1995).*
 [e] *Berthold, Nilsson & Rydmark (1983).*
 [f] *Ruff et al. (2013).*
 [g] *Chomiak & Hu (2009).*
 [h] *Sadik et al. (2011).*

Lamella are produced by many layers of processes from oligodendrocytes with significant membrane resistivity (*Bakiri et al., 2011*). The resistance of the myelin was scaled linearly by the number (*n*) of lamella (*Chomiak & Hu, 2009*). The standard electric intensity was 200 V/m. The maximum intensity is the one that can cause membrane electroporation (*Sadik et al., 2011*).

### Software packages

Equations were derived with Mathematica 10 (Wolfram Research, Inc. Champaign, IL, USA). Numerical simulations were performed with Matlab 8.4.0 (The MathWorks, Inc. Natick, MA, USA).

## RESULTS

### Analytical expressions of axonal transmembrane potential (*Vm*) and voltage drop on the myelin sheath (φ) under transverse electric stimulation

The solution for Laplace's equation (Eq. (3)) was written in the form (*Griffiths, 1999*)

$$V(r,\theta) = A_0 \ln(r) + B_0 + \sum_{n=1}^{\infty} r^n [A_n \sin(n\theta) + B_n \cos(n\theta)] + \sum_{n=-\infty}^{-1} r^n [C_n \sin(n\theta) + D_n \cos(n\theta)]$$

(8)

The expression was further simplified for the five modeled regions (*Griffiths, 1999*; *Ye et al., 2011*)

$$V_n = \left(\frac{A_n}{r} + C_n r\right)\sin\theta \tag{9}$$

where $A_n$, $C_n$ were unknown coefficients ($n = 0, 1, 2, 3, 4$). These coefficients were solved in the Appendix (File S1), by considering boundary conditions (A–D). Substituting $A_3$, $C_3$ into (9), we obtained the expression of voltage inside the axolemma

$$V_3 = -\frac{8a^2 b^2 c^2 E_0 S_0 S_1 S_2 \cos\theta}{r}\left[\frac{\text{term1}}{\text{term2} + \text{term3}}\right]. \tag{10}$$

where

$\text{term1} = d^2(S_3 - S_4) + r^2(S_3 + S_4)$

$\text{term2} = b^2(S_0 - S_1) \times \{c^2(S_1 + S_2)[d^2(S_2 + S_3)(S_3 - S_4) + c^2(S_2 - S_3)(S_3 + S_4)]$
$\quad + b^2(S_1 - S_2)[d^2(S_2 - S_3)(S_3 - S_4) + c^2(S_2 + S_3)(S_3 + S_4)]\}$

$\text{term3} = a^2(S_0 + S_1) \times \{c^2(S_1 - S_2)[d^2(S_2 + S_3)(S_3 - S_4) + c^2(S_2 - S_3)(S_3 + S_4)]$
$\quad + b^2(S_1 + S_2)[d^2(S_2 - S_3)(S_3 - S_4) + c^2(S_2 + S_3)(S_3 + S_4)]\}$

The axonal transmembrane potential (*Vm*) of the field was obtained by subtracting the membrane potential at the inner surface from that of the outer surface of the axon (*Kotnik, Bobanovic & Miklavcic, 1997*; *Kotnik & Miklavcic, 2000a*; *Ye et al., 2010, 2011*), $V_m = V_3(r = d) - V_3(r = c)$. For a myelin-covered axon (File S2),

$$V_m = 8a^2 b^2 c(c - d)E_0 S_0\left[\frac{\text{term4}}{\text{term2} + \text{term3}}\right]\cos\theta \tag{11}$$

Where

$\text{term4} = S_1 S_2[d(-S_3 + S_4) + c(S_3 + S_4)]$

Voltage drop ($\Phi$) across the myelin sheath was obtained by subtracting the myelin potential at the inner surface from the outer surface of the myelin (File S2)

$$\varnothing = 2a(a - b)E_0 S_0\left[\frac{\text{term5} + \text{term6}}{\text{term2} + \text{term3}}\right]\cos\theta \tag{12}$$

Where

$\text{term5} = b\{c^2(S_1 + S_2) \times [d^2(S_2 + S_3)(-S_3 + S_4) - c^2(S_2 - S_3)(S_3 + S_4)]$
$\quad - b^2(S_1 - S_2)[d^2(S_2 - S_3)(S_3 - S_4) + c^2(S_2 + S_3)(S_3 + S_4)]\}$

$\text{term6} = a\{c^2(S_1 - S_2) \times [d^2(S_2 + S_3)(S_3 - S_4) + c^2(S_2 - S_3)(S_3 + S_4)]$
$\quad + b^2(S_1 + S_2)[d^2(S_2 - S_3)(S_3 - S_4) + c^2(S_2 + S_3)(S_3 + S_4)]\}$

*Vm* and $\Phi$ were functions of both field properties and tissue properties. The field properties included the orientation of the field and its intensity. The tissue properties include the electric parameters (conductivity and di-electricity) and the geometrical parameters (i.e., diameters of the axon). The above *Vm* expression for the myelin-covered axon (Eq. (11)) was further simplified for a bare axon by assuming $S_1 = S_0$ and $S_2 = S_0$ (File S3),

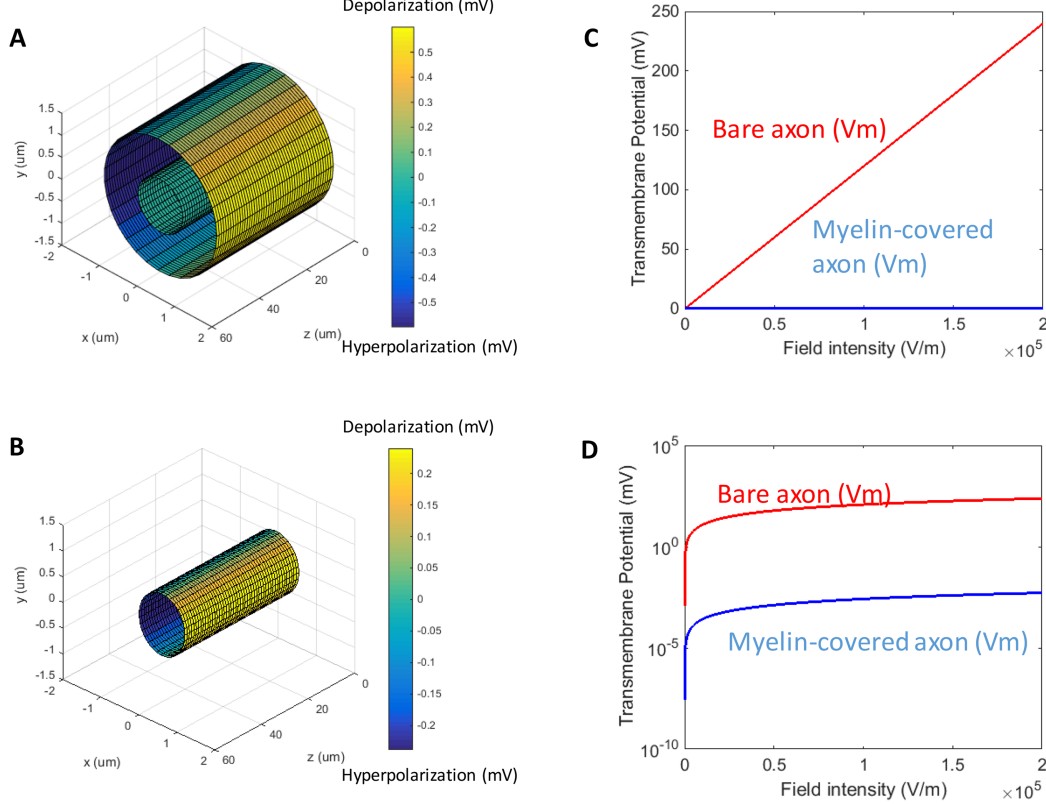

**Figure 2 Polarization of a myelin-covered axon (A) and a bare axon (B) in a transverse electric field.** The $Vm$ was calculated by Eqs. (11) and (13), for the myelin-covered axon and the bare axon, respectively. $\phi$ was calculated by Eq. (12). All calculations were based on the standard parameters in Table 1. The color maps represented the amount of polarization (in mV). (C) Effect of transverse electric field intensity on axonal polarization in myelin-covered and bare axons. (D) Log plot of (C).

$$V_m = \frac{2c(c-d)E_0 S_0 [d(-S_3 + S_4) + c(S_3 + S_4)]}{d^2(S_0 - S_3)(S_3 - S_4) + c^2(S_0 + S_3)(S_3 + S_4)} \cos \theta \tag{13}$$

## Impact of electric field properties on *Vm*

When a transverse electric field penetrates the axolemma, the geometrical pattern of *Vm* is determined by the axon's orientation to the electric field. The axolemma should be hyperpolarized wherever an electric current enters the membrane and be depolarized wherever the current extrudes from the membrane (*Ye & Steiger, 2015*). We plotted the transmembrane potential for a 50 μm myelinated axon (Fig. 2A) and a straight bare axon (Fig. 2B), based on the calculated *Vm* using standard values (Table 1). As expected, the locations of maximum polarization were at two lines corresponding to when θ = 180° (hyperpolarization, blue) and θ = 0° (depolarization, yellow), respectively. The axons were not polarized at the locations where θ = 90° and θ = 270°.

When the axon was wrapped by a thick myelin sheath, the geometrical pattern of the axolemma depolarization (Fig. 2A) remained identical to an unmyelinated axon
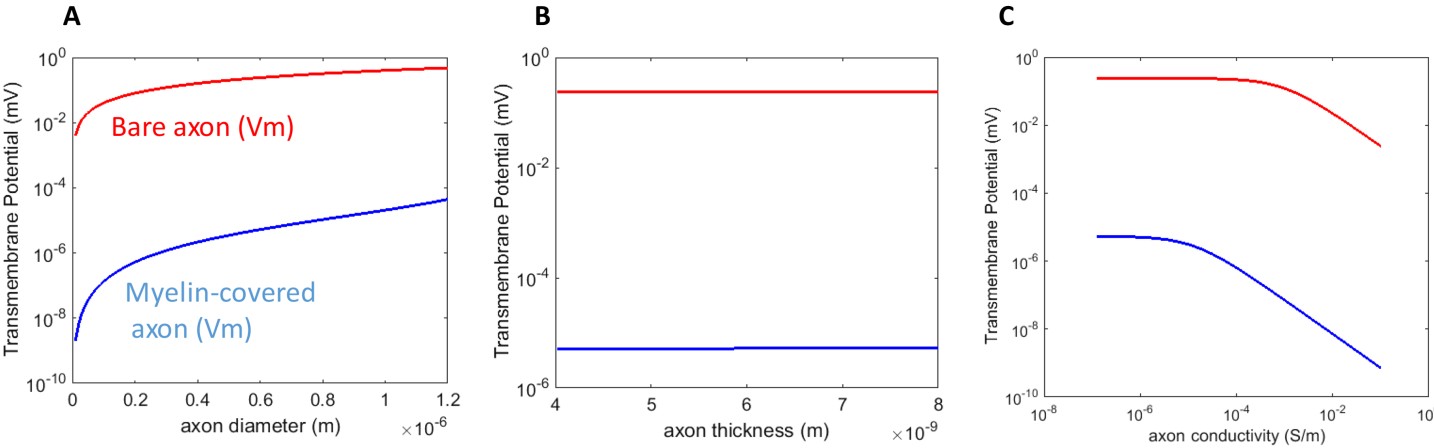

**Figure 3 Dependency of *Vm* on the biophysics properties of the axon.** (A) Axolemma diameter. (B) Axolemma thickness. (C) Axolemma conductivity.

(Fig. 2B). However, since a large voltage gradient (ϕ) was generated on the myelin sheath rather than on the axolemma, *Vm* was quantitatively negligible for the myelinated axon. With the standard values in Table 1, the maximum Φ was 0.6 mV for the myelin sheath, and the maximum *Vm* was only $0.53 \times 10^{-2}$ μV for the axolemma. In contrast, when the axon was not myelinated, the maximum *Vm* was 0.24 mV.

For both the myelin-covered and bare axons, *Vm* was proportional to the intensity of the electric field (Figs. 2C and 2D). Two kV/cm was sufficient in inducing electroporation (*Sadik et al., 2011*). This intensity induced a *Vm* of 5.3 μV for the myelin-covered axon. For a bare axon, it induced a *Vm* of 239.9 mV, which is sufficient to break down the structure of the membrane (*Gehl, 2003*; *Kinosita & Tsong, 1977*). These results suggest that the myelin sheath could provide a "shielding effect" on the axolemma against field-induced excessive polarization and structure disruption.

## Impact of axonal properties on *Vm*

We investigated the dependency of *Vm* on the axonal properties, including the geometrical features (axon radius and membrane thickness) of the axon, and its conductivity. For the parametric analysis, we plotted the maximum polarization (θ = 0° on the axon surface, Figs. 1 and 2) when one parameter was varied through its defined value range, while the others were maintained at their standard values.

An axon with a larger radius was associated with a greater *Vm* for both the myelin-covered axon and bare axon (Fig. 3A) under a transverse field stimulation. Axon thickness, however, did not significantly affect *Vm* (Fig. 3B). *Vm* was insensitive to the axonal conductance changes within its physiological range ($10^{-8}$–$10^{-6}$ S/m), in agreement with the literature that studied spherical cell polarization in an electric field (*Kotnik, Bobanovic & Miklavcic, 1997*; *Kotnik & Miklavcic, 2006*). However, when axolemma conductivity was significantly increased (>$10^{-3}$ S/m) due to membrane disruption and leakage, such as during electroporation (*Mossop et al., 2004, 2007*),

axolemma depolarization decreased significantly for both the myelin-covered and bare axons (Fig. 3C).

### Impact of demyelination on axonal *Vm*

Myelin, like a neuronal cell membrane, is constructed of a lipid bilayer that contains a hydrophobic center and hydrophilic surface. Myelin wraps around an axon numerous times, each layer acting like multiple resistors in series. Demyelination occurs in many neurological diseases such as spinal cord injury (*Ye et al., 2012*), cerebral palsy (*Ruff et al., 2013*) and multiple sclerosis (*Lazzarini, 2004*). Demyelination is defined by the significant loss of myelin thickness (*Mainero et al., 2015*; *Manogaran et al., 2016*) and increased conductivity of the myelin.

We first studied how the loss of myelin layers could affect *Vm* in a transverse electric field. We systematically decreased the myelin thickness from 4.0–0.1 μm. The conductivity of the myelin increased linearly with the reduction of the myelin thickness. This caused a reduction in the potential drop across the myelin sheath, but it did not significantly affect axonal depolarization (Fig. 4). The transverse electric field was ineffective in inducing axonal depolarization, assuming the remaining myelin sheath could maintain low conductivity ($\sim 10^{-7}$ S/m).

We then investigated how an increase of myelin conductivity could affect *Vm* in a transverse electric field. When myelin conductivity was as low as $5 \times 10^{-5}$ S/m, reduction in myelin thickness did not lead to dramatic changes in *Vm* (Fig. 5A). Instead, it led to a voltage drop across the myelin sheath ($\Phi$). When myelin conductivity was increased to $5 \times 10^{-3}$ S/m, *Vm* could exceed $\phi$ for an extremely thin myelin sheath (Fig. 5B). For a very leaky myelin (myelin conductivity is $5 \times 10^{-1}$ S/m), the axon could be significantly depolarized at any myelin thickness (Fig. 5C), and *Vm* could be greater than $\Phi$ for a thin myelin sheath (Fig. 5D). However, $\Phi$ still dominated for axons with thick myelin sheaths (Fig. 5E). In conclusion, demyelination could cause a re-distribution of the potentials between the axolemma and myelin under transverse electric stimulation. Increases in myelin conductivity during demyelination could cause the voltage distribution to shift from the myelin sheath to the axon. Axonal depolarization became prominent when significant reduction of myelin conductivity occurred during demyelination.

## DISCUSSION

This work provides a novel analytical expression that describes the membrane polarization of axons (myelinated and bare) under transverse field stimulation. It analyzes the biophysical factors that affect axonal polarization under physiological conditions, and under pathological conditions such as demyelination. Finally, it provides the needed term to modify the current cable equation, so that the equation can account for the effects of more realistic field, which include both the transverse and parallel directions.

### Impact of field orientation on *Vm* in transverse current stimulation

The model shows that the longitudinal axon was polarized with a distinct geometrical pattern by the transverse electric field, which was dependent on the orientation of the axon

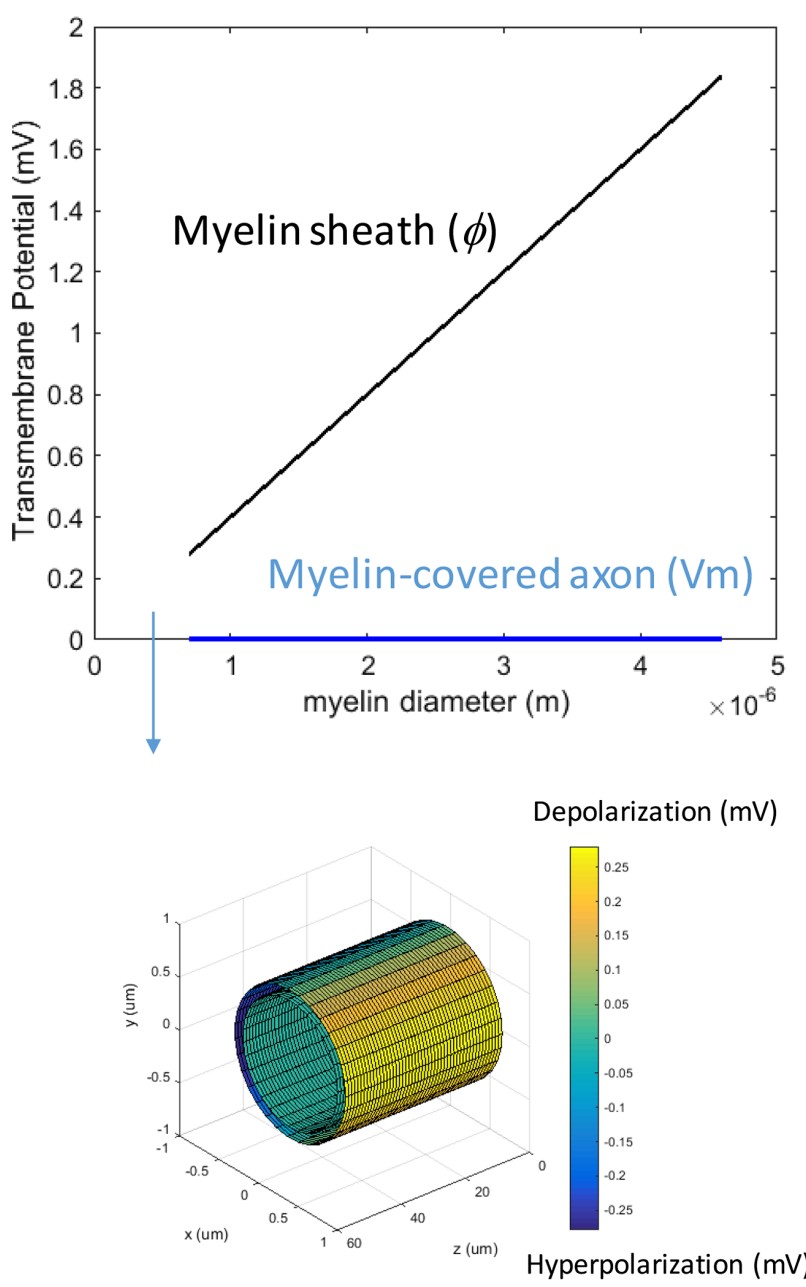

**Figure 4 Effects of decreased myelin thickness on axonal polarization.** Reduction of myelin thickness from 3.4 to 0.1 μm (and linear increase of its conductivity) caused a significant reduction in ϕ, but not *Vm*. For the inset example, axon diameter = 0.6 μm. Myelin thickness = 0.1 μm.

in the field. Previously, regional polarization has been observed in a variety of modeling and experimental studies for cells under electric (*Durand, 2003*; *Lee & Grill, 2005*; *Lu et al., 2008*; *Teruel & Meyer, 1997*) and magnetic field stimulation (*Schnabel & Struijk, 1999*, *2001*; *Ye, Cotic & Carlen, 2007*). Functionally, orientation of the electric field to the axon is important for the excitation of axons, such as those from the retina ganglion cells (*Grumet, Wyatt & Rizzo, 2000*). Since only a small patch of membrane is depolarized

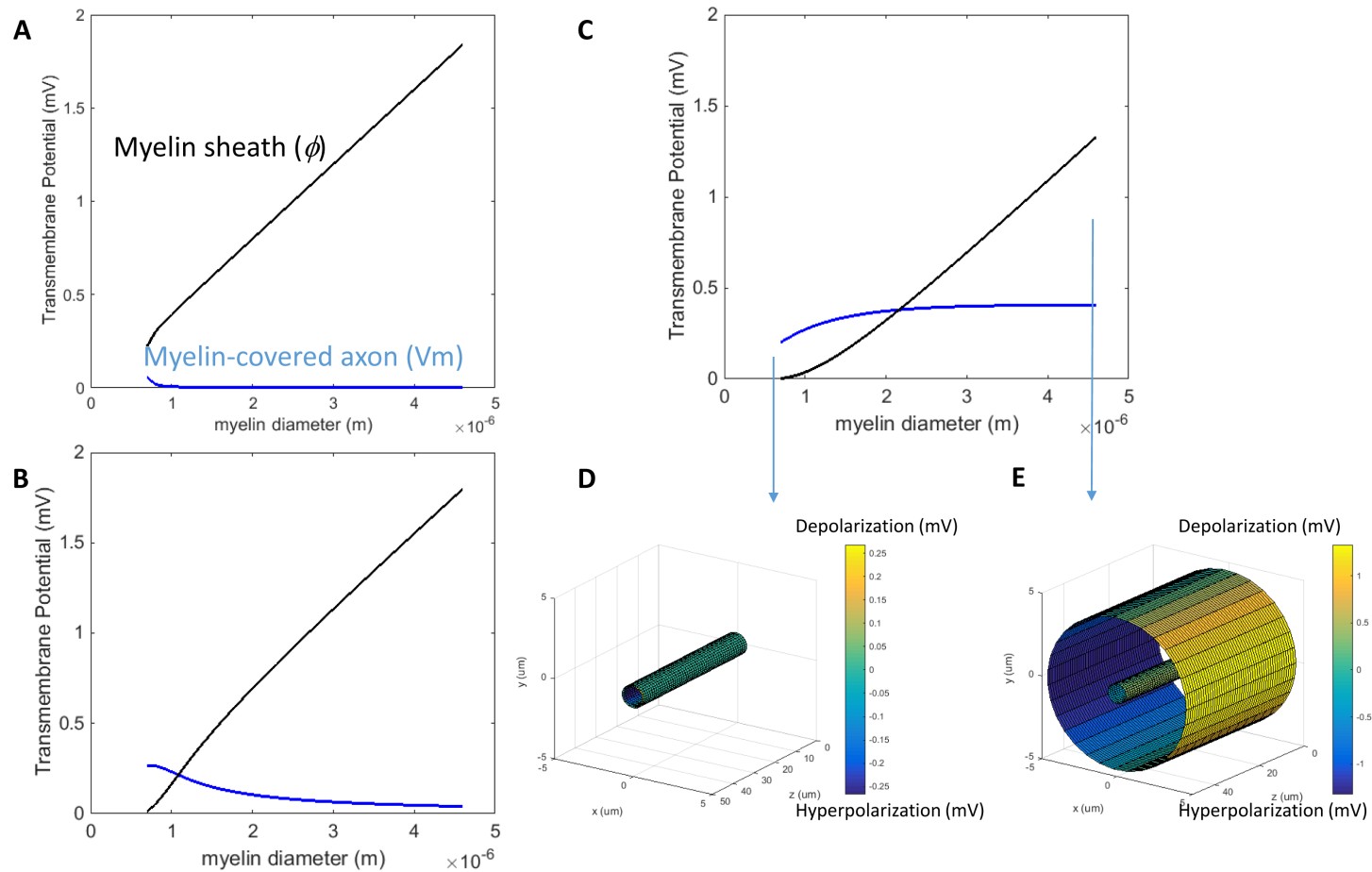

**Figure 5 Effects of a leaky myelin sheath on axolemma polarization in a transverse electric field.** Conductivity of each myelin layer was (A) $5 \times 10^{-5}$ S/m, (B) $5 \times 10^{-3}$ S/m, and (C) $5 \times 10^{-1}$ S/m, respectively. (D) Example of axon polarization when axon diameter = 0.6 μm and myelin diameter = 0.7 μm. (E) Example of axon polarization when axon diameter = 0.6 μm and myelin diameter = 4.6 μm.

in the transverse field, it is reasonable to speculate that voltage gated ion channels may have a diverse response to the field, depending on their location on the membrane patch. This could cause the threshold for activation to be higher than that observed from axons in longitudinal fields that induce the same peak depolarization (*Schnabel & Struijk, 2001*). The higher threshold may explain the relative poor efficiency of axonal activation by transverse field stimulation (*McNeal, 1976*; *Ranck, 1975*).

## Impact of axon's biophysical properties on *Vm*

We found that *Vm* was dependent on the intrinsic tissue properties of an axon. We observed that *Vm* was greater in larger diameter axons than in smaller ones (Fig. 3A). This observation is in agreement with the notion that larger diameter axons are associated with lower excitation thresholds (*Basser & Roth, 1991*; *Carbunaru & Durand, 1997*; *Garnsworthy et al., 1988*; *Reilly, 1989*). Selective activation of different size fibers has significant clinical implications, such as pain relief (*Meyerson & Linderoth, 2000*), which can be achieved by novel design of the electric field (*Konings, 2007*). In deep brain

stimulation, the effects of the electric currents within different brain regions were dependent on the fiber sizes (*Sotiropoulos & Steinmetz, 2007*). In addition, an increase in axolemma conductivity decreases the axon's sensitivity (buildup of $Vm$) to the transverse field (Fig. 3C), suggesting a shunting effect to the transverse current. In conclusion, the effectiveness of transverse stimulation relies on the physiological features of the target axon.

Axonal diameter could change under certain pathological situations. For example, axon swelling occurs during focal demyelination (*Kolaric et al., 2013*), as a consequence of aglycemia (*Allen et al., 2006*), anoxia (*Waxman et al., 1992*) or ischemia (*Garthwaite et al., 1999*). It is speculated that these pathological changes could potentially render the enlarged axons more sensitive to the transverse electric field.

### Impact of demyelination and other pathological conditions on *Vm* in transverse electric stimulation

Dynamic changes of myelin occur during demyelination. It is unknown if pathological demyelination could affect the sensitivity of a myelinated axon to a transverse electric field. While myelin-covered axons could only be slightly depolarized by the transverse field, bare axons can have a moderate buildup of $Vm$ (Fig. 2), especially when the axon diameter is large (Fig. 5A). We used the model to test two possibilities of reduced myelination and their impacts on $Vm$. Reduction in the myelin thickness, along with a scaled linear increase in myelin conductance, was not sufficient to enhance depolarization (Fig. 4). In contrast, $Vm$ was enhanced when the myelin sheath became electrically leaky (highly conductive) (Fig. 5). It is therefore expected that transverse electric fields could apply variable axonal depolarization, depending on the myelin conductivity changes during the process of demyelination. Electrical stimulation protocols for the treatment of demyelination diseases (*Dooley & Sharkey, 1981*; *Dooley et al., 1978*) could be further optimized by considering remyelination/demyelination factors during treatment, to ensure maximum outcomes.

Dynamic changes of myelin also occur during development (*Sturrock, 1980*), neural regeneration (*Huang et al., 2013*), and pathological situations such as traumatic brain injury (*Robain & Mandel, 1974*; *Tyler, 2012*). At the cellular level, membrane resistance of the oligodendrocyte could change during development and maturity (*Karadottir et al., 2005*), and in a medium with low osmolarity (*Kimelberg & Kettenmann, 1990*). It is speculated that these dynamic changes in the myelin properties could cause the axons to react differently to the electric field. This supports the notion that the dynamic interaction between the electric field and the neuronal tissue, as well as the outcome of the stimulation, are determined by both the electric parameters and the tissue properties (*Ye & Steiger, 2015*).

### Modification of the cable equation to include the transverse field

The analytical expression of the $Vm$ term could potentially be used to modify the current cable equation to account for both $E_{//}$ and $E_\perp$. *Ruohonen et al. (1996)* modified the cable equation to be in the form of $\lambda^2 \frac{\partial^2 \phi_m}{\partial x^2} - \tau \frac{\partial \phi_m}{\partial t} - \phi_m - 2c(\alpha E'_{//} - E_\perp) = 0$.

Here, $\alpha E'_{//} - E_\perp$ is interpreted as the *modified activating function*, where $\alpha = \frac{\lambda^2}{2c} = \frac{R_m}{4R_i}$ (*Ruohonen et al., 1996*). Comparatively large values of $\alpha$ indicates that $E'_{//}$ is responsible for the majority of excitation, while $\alpha = 0$ indicates that $E_\perp$ is more important.

In this modified equation, the term $2cE_\perp$ is the membrane potential created by the transverse field. For magnetic stimulation, an analytical expression (*Ye et al., 2011*) is available to replace this term for the modified cable equation. For direct electric stimulation, the $Vm$ term for the unmyelinated axon (Eq. (13)) can be used to replace the $2cE\perp$ term, to include the impact of the transverse electric field. Previously, effects of the transverse field and the axial field have been compared in several works (*Lontis, Nielsen & Struijk, 2009*; *Ruohonen et al., 1996*; *Yu, Zheng & Wang, 2005*). The transverse electric field is required to be several times greater than the longitudinal field to produce comparable results (*McNeal, 1976*; *Ranck, 1975*; *Ruohonen et al., 1996*). A precondition for the modified activation function to yield accurate results is for the electric field to be approximately uniform and be perpendicular to the axon fiber (*Schnabel & Struijk, 2001*), which is readily satisfied in our model.

## Limitations and future directions

This paper was not intended to fully elucidate the mechanisms behind transverse field activation of nerve tissue, since it did not include any ionic channel mechanisms. The model also does not necessarily apply to the stimulation of fiber bundles. Axons within a bundle could interfere with other axon's polarization under a transverse electric field (*Pourtaheri et al., 2009*). Local electric fields could be perturbed by an axon, which produces a small, secondary effect on the surrounding axons (*Lee & Grill, 2005*; *Susil, Semrov & Miklavcic, 1998*). In a nerve bundle, $Vm$ could also be a function of the anisotropy of the bundle (*Nagarajan & Durand, 1995*), which was not studied in the present model. Finally, the transverse field could be significantly weaker due to the lower values of conductance of surrounding perineurium (*Struijk & Schnabel, 2001*). More complicated modeling work should resort to numerical methods, whose accuracy can be validated by the analytical results from this work.

The model predicts that the node section in a myelinated axon will have the same polarization as the unmyelinated axon. If one considers that the node has a much higher density of Na$^+$ channel distribution (*Freeman et al., 2016*), it is predicted that myelinated axons will have a lower threshold of activation under transverse electric field. This model prediction could be tested by stimulating a structure that contains both unmyelinated and myelinated axons, such as the corpus callosum (*Crawford, Mangiardi & Tiwari-Woodruff, 2009*; *Ruff et al., 2013*). With the strong stimulus being applied on both type of axons (*Ruff et al., 2013*), action potentials should be triggered first in the myelinated axons.

## CONCLUSIONS

This work provides novel analytical expressions of the electrically-induced transmembrane potential ($Vm$) for a myelin-covered axon and a bare, unmyelinated axon, under a transverse DC electric field. Results show that the myelin sheath shields the axon from extensive depolarization. Demyelination could alter axon's sensitivity to a transverse electric field if the process of demyelination involves significant increases in the electric

conductance of the myelin. The analytical solution of $Vm$ for the unmyelinated axon can be used to improve the activation function of the current cable equation that describes electric stimulation.

## APPENDIX—DETERMINING UNKNOWN COEFFICIENTS $A_n$, $C_n$ IN EQ. (9) USING BOUNDARY CONDITIONS (A–D)

At an infinite distance, according to boundary condition (C), $V_o = -E_0 r \cos \theta$. Therefore, $a_0 = -E_0$. Since $V$ was bounded at $r = 0$ (boundary condition D), $C_4 = 0$.

Expressions for the potential distribution in the five modeled regions were:

$$V_0 = -E_0 r \cos \theta + \frac{C_0}{r} \cos \theta \tag{A-1}$$

$$V_1 = A_1 r \cos \theta + \frac{C_1}{r} \cos \theta \tag{A-2}$$

$$V_2 = A_2 r \cos \theta + \frac{C_2}{r} \cos \theta \tag{A-3}$$

$$V_3 = A_3 r \cos \theta + \frac{C_3}{r} \cos \theta \tag{A-4}$$

$$V_4 = A_4 r \cos \theta \tag{A-5}$$

The $\vec{r}$ components of $\nabla V$ (from Eq. (1)) were continuous across the interfaces (boundary condition A), and the normal components of the current density were continuous across the interfaces (boundary condition B). These boundary conditions yield the following set of equations:

On the #0#1 interface ($r = a$)

$$-E_0 a + \frac{C_0}{a} = a A_1 + \frac{C_1}{a} \tag{A-6}$$

$$S_0 \left( -E_0 - \frac{C_0}{a^2} \right) = S_1 \left( A_1 - \frac{C_{11}}{a^2} \right) \tag{A-7}$$

On the #1#2 interface ($r = b$)

$$b A_1 + \frac{C_1}{b} = b A_2 + \frac{C_2}{b} \tag{A-8}$$

$$S_1 \left( A_1 - \frac{C_1}{b^2} \right) = S_2 \left( A_2 - \frac{C_2}{b^2} \right) \tag{A-9}$$

On the #2#3 interface ($r = c$)

$$c A_2 + \frac{C_2}{c} = c A_3 + \frac{C_3}{c} \tag{A-10}$$

$$S_2 \left( A_2 - \frac{C_2}{c^2} \right) = S_3 \left( A_3 - \frac{C_3}{c^2} \right) \tag{A-11}$$

On the #3#4 interface ($r = d$)

$$dA_3 + \frac{C_3}{d} = dA_4 \qquad (A\text{-}12)$$

$$S_3\left(A_3 - \frac{C_3}{d^2}\right) = S_4 A_4 \qquad (A\text{-}13)$$

We solved (A-6) to (A-13) to obtain the unknown coefficients (File S1). These coefficients will be substituted into (A-1) to (A-5) to obtain the analytical expression of the voltages in the five regions (File S1).

## LIST OF ABBREVIATIONS

| | |
|---|---|
| **DC** | Direct current |
| $E_0$ | Intensity of the externally applied DC electric field (V/m) |
| $Vm$ | Transmembrane potential induced by the DC electric field across the axolemma (mV) |
| $\phi$ | Potential drop across the myelin sheath (mV) |
| $E_{//}$ | Electric field that is parallel to the axon |
| $E_\perp$ | Electric field that is perpendicular (traversal) to the axon. |

## ACKNOWLEDGEMENTS

The authors thanks Austen Curcuru for the assistance in deriving the equations.

### Funding

The authors received no funding for this work.

### Competing Interests

The authors declare that they have no competing interests.

### Author Contributions

- Hui Ye conceived and designed the experiments, performed the experiments, analyzed the data, contributed reagents/materials/analysis tools, prepared figures and/or tables, authored or reviewed drafts of the paper, approved the final draft.
- Jeffrey Ng authored or reviewed drafts of the paper, approved the final draft, paper revision.

### Data Availability

The raw data are provided in the Supplemental Files.

### Supplemental Information

Supplemental information for this article can be found online at http://dx.doi.org/10.7717/peerj.6020#supplemental-information.

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
