# Peer review of "Shielding effects of myelin sheath on axolemma depolarization under transverse electric field stimulation"

_PeerJ, doi:10.7717/peerj.6020_

## Round 0.1 · original submission · Major Revisions

Both reviewers commented that the equations need to be reformatted to make them more accessible. The authors should also address their questions regarding the significance of the presented findings and how these theoretical derivations should be interpreted in experimental settings.

Reviewer 1 ·

Basic reporting

The paper was nicely written, but the mathematical equations were not well formatted and not consistent in font types:
Eq (1): Is there supposed to be an arrow (instead of a box) on top of the electric field symbol E?
Eq (3): Is there a partial derivative symbol missing?
Eq (8): ln, sin and cos, missing brackets, or non-italic.
Eq (10 & 13): cos should be round bracket
Eq(10): missing bracket
Eqs for term1,2,3,4: change of font size from main content, suggest move to appendix or footnote
Eqs for coefficients in appendix: unreadable, should be broken down at some point and organised neatly

Experimental design

The authors in this paper were trying to prove mathematically the capability of exciting neural axons with a transverse electric field, and to identify the influence of axonal geometric parameters on the excitation.
The paper in itself was nicely written and the deduction was good. The findings however were rather unsurprising, for the existence of the electric field, regardless of transverse or parallel, can result in local polarity change with the axon, which in turn causes voltage difference and may lead to substantial transmembrane voltage drop and result in depolarisation.
Due to the irregular trajectories of nerve fibres, it is rather difficult to pin-point whether the electric field is parallel or perpendicular to the fibres, or even at the particular segments; and chances are that both types should be present at the same time. Therefore, a more important/interesting question is, when a parallel electric field is present, is transverse electric field still important?

Validity of the findings

See comments to 2nd point.

·

Basic reporting

Minor issue: many of the equations presented are multi-term, complex mathematical expressions that appear to be parsed from a symbolic math toolbox - it would greatly improve the presentation if these were formatted in a professional math typesetter such as Latex or Word equation editor. This applies to lines 152-154, 162-164, 169-172 and 353-389.

Other typographical errors:
Line 99: in equation 1, the overscript for E seems incorrect - should it be an arrow to denote a vector?
Line 211: replace “did not significant affect” with “did not significantly affect”.
Line 220: replace “constructs of a lipid bilayer” with “is constructed of a lipid bilayer”.
Lines 420-421: Delete "[pii]" and format reference correctly (doi is on a separate line)
Lines 484-486: As above, delete "[pii]" and format reference correctly.
Line 496: Delete "Academic".
Lines 502-504: Delete "[pii]" and format reference correctly.
Lines 580-583: Delete "[pii]" and format reference correctly.

Experimental design

This paper presents theoretical derivations of transverse electrc field stimulation of myelinated axons, finding that the myelin sheath provides a significant resistance in this mode, effectively shielding the axon from activation. Although the mathematical derivations are interesting, the significance of this finding is hard to justify, particularly given that myelinated axons have closley-spaced unmyelinated nodes of Ranvier, and any applied transverse fields are likely to stimulus these first. Of more interest is the effect of transverse stimulation in this more realistic scenario, where the shielding effect of myelin is likely to be vastly diminished - is the stimulus threshold actually lower when compared to the unmyelinated axon? This would have been far more interesting and useful to explore theoretically.

Validity of the findings

Derivations are intersting - I have not seen these explicitly published before. However, significance of this work is not clear, as per my comments above.

Additional comments

See my comments above.

---

## Round 0.2 · Major Revisions

While the revised manuscript has improved considerable the reviewers have expressed continued concern about the clarity of the presented equations. Moreover, questions were raised whether the high stimulation intensity that are used are realistic parameter settings for electrical stimulations.

Reviewer 1 ·

Basic reporting

1. The equations between line 149 and 182 are confusing. Suggest a better formatting, such as everything belonging to the same equation stays to the right of the '=' sign, even after a line break.

2. Unable to find the paper Pourtaheri 2008. Citation information incomplete.

Experimental design

Despite the authors' best efforts, I still find they have not yet addressed to all the comments from the previous reviewers. For the foremost, the impact of the study.

The transverse EF may be able to activate the cells, however, under what condition? The value of EF the authors used for their study is 20k-200k V/m, which they got from a study of electroporation, and thus achieved a depolarisation in a micron range. However, it's hard to imagine a neural stimulation in an ordinary situation using such a high stimulation intensity without damaging the nerve tissue. In terms of brain stimulation, the electro-shock therapy is the one using a relatively high stimulation, and the EF normally peaks at 200 V/m, resulting in significant side effects. Therefore, such an unrealistic value does not support the findings of this study. On the contrary, it suggested that only in very extreme and rare case, the transverse EF can cause depolarisation.

In addition, it is still debatable whether in the literature the authors referenced in this paper the transverse EF is the key player in exciting the nerves. For instance in cuff/cylindrical electrode, the excitation may be due to the parallel EF drop at the boundary of the electrode; and in the MS situation, it may be due to the uneven shape/trajectory of the nerves... none of these definitively supports that it is the transverse EF but not the parallel EF that plays a role in the excitation.

Indeed, 'with significant transverse stimuli, the cable model assumptions are known to be invalid', and I like the idea of modifying the cable eq to include transverse EF to make it more precise. However, the authors have failed to provide us with the necessity to do so.

Validity of the findings

Please see my comments above.

·

Basic reporting

This revised manuscript represents a substantial improvement from the previous submission. Issues I had raised earlier have largely been addressed in this revision, although the presentation of some of the more complex equations in the paper could still be improved, as per my suggestions below:
Line 156, Eq (10): Place large brackets around the final term [term1/(term2 + term3)].
Lines 159-161: This is a complex equation broken over several lines. It would improve readability by inserting an explicit multiplication operator x after the first term on line 159. Also line 161 appears to have an extraneous single quote, which could be misinterpreted as a differentiation operator.
Lines 162-162. As above, insert an explicit multiplication operator 'x' after the first term on line 162.
Line 170, Eq (11): Place large brackets around the term [term4/(term2 + term3)]
Line 176, Eq (12): Place large brackets around the term [(term5 + term6)/(term2 + term3)]
Lines 178-179. Insert an explicit multiplication operator 'x' after the first term on line 178.
Lines 180-181. Insert an explicit multiplication operator 'x' after the first term on line 180.

Other typographical errors are:
Line 15: Replace "an effect method" with "an effective method".
Line 19: Replace "electric filed" with "electric field".
Line 25: Replace "Polarization of both axons were" with "Polarization of both axons was".
Line 69: Replace "with is considered" with "which is considered".
Line 104: Replace "Inside the cylindrical coordinates" with "Using the cylindrical coordinates".

Experimental design

The authors have addressed my earlier comments on experimental design.

Validity of the findings

Derivations of transverse electric field contribution to axonal depolarization are novel and interesting. The authors have also addressed my initial concerns as to the significance of their findings.

Additional comments

As per above, my only recommendations are that the authors address improve presentation of their complex equations and correct some remaining typographical errors.

---

## Round 0.3 · accepted · Accept

The authors have made several changes to the manuscript in response the second round of reviews. They have reviewed the literature on transversal field for membrane polarization in more detail in the introduction and corrected the errors in the equations. These revisions adequately addressed the remaining comments.

#